# SPARSE TRAINING OF DISCRETE DIFFUSION MODELS FOR GRAPH GENERATION

## ABSTRACT

Generative models for graphs often encounter scalability challenges due to the inherent need to predict interactions for every node pair. Despite the sparsity often exhibited by real-world graphs, the unpredictable sparsity pattern of their adjacency matrices, stemming from their unordered nature, leads to quadratic computational complexity. In this work, we introduce SparseDiff, a denoising diffusion model for graph generation that can exploit sparsity during its training phase. At the core of SparseDiff is a message-passing neural network tailored to predict only a subset of edges during each forward pass. When combined with a sparsity-preserving noise model, this model can efficiently work with edge list representations of graphs, paving the way for scalability to much larger structures. During the sampling phase, SparseDiff iteratively populates the adjacency matrix from its prior state, ensuring the prediction of the full graph while controlling memory utilization. Experimental results show that SparseDiff simultaneously matches state-of-the-art generation performance on both small and large graphs, highlighting the versatility of our method. [1]

## 1 INTRODUCTION

Random graph models have been foundational in graph generation, with a rich legacy spanning several decades (Erdős et al., 1960; Aiello et al., 2000; Barabási, 2013). However, recent interest has gravitated towards learned graph models, primarily due to their enhanced ability to represent intricate data distributions. Traditional frameworks like generative adversarial networks (De Cao & Kipf, 2018) and variational autoencoders (Simonovsky & Komodakis, 2018) predominantly addressed graphs with a maximum of 9 nodes. This limitation was somewhat alleviated with the advent of denoising diffusion models (Niu et al., 2020; Jo et al., 2022; Vignac et al., 2023a), elevating capacity to roughly 100 nodes. However, these models are still not scaled for broader applications like transportation (Rong et al., 2023) or financial system anomaly detection (Li et al., 2023).

The primary bottleneck of many generative graph models is their computational complexity. While many natural graphs are sparse, the unordered nature of graphs makes it challenging to exploit this trait. Without a predetermined sparsity pattern, models frequently make exhaustive predictions for every node pair, constraining them to a ceiling of ∼200 nodes (Vignac et al., 2023a). Proposed methods to circumvent this issue include imposing a node ordering (Dai et al., 2020), assembling sub-graphs (Limnios et al., 2023), generating hierarchically (Karami, 2023; Jang et al., 2023), and conditioning the generation on a sampled degree distribution (Chen et al., 2023). These methods, designed for large graphs, implicitly make assumptions about the data distribution which sometimes reflects a poor ability to model very constrained graphs such as molecules (Chen et al., 2023; Kong et al., 2023).

To address these limitations, we propose SparseDiff, a generative model for graphs that exploits sparsity in its training phase by adopting edge list representations. Unlike other scalable models, SparseDiff leverages the intrinsic sparsity of training graphs without requiring additional assumptions about the data distribution. SparseDiff defines a sparse diffusion model that comprises three primary components: 1). A noise model designed to retain sparsity throughout the diffusion process; 2). A loss function computed on a set of random node pairs; 3) A sparse graph transformer rooted

---

[1]Our code is available at https://anonymous.4open.science/r/SparseDiff-B861.

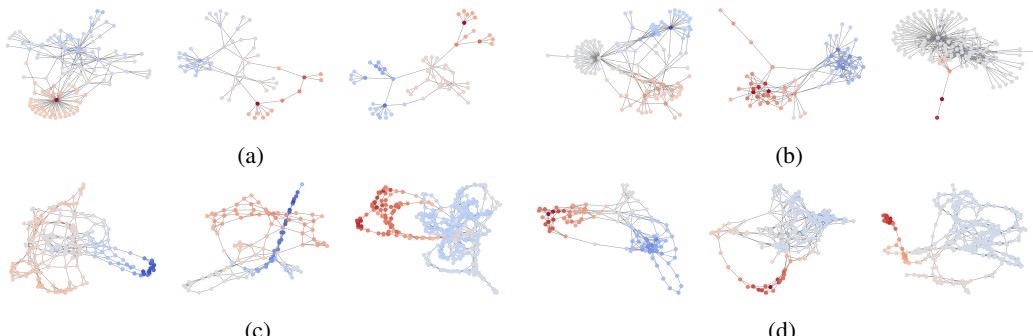

Figure 1: Samples from SparseDiff trained on large graphs. (a) Ego training set (50 to 399 nodes); (b) Generated Ego graphs; (c) Protein training set (100 to 500 nodes); (d) Generated Protein graphs.

in the message-passing framework. During the sampling process, our model iterates over all pairs of nodes and progressively builds the predicted graph.

Experiment demonstrates that, despite its simplicity, SparseDiff achieves generation performance comparable to scalable models on large graphs. Additionally, it attains results similar to state-of-the-art dense models on small molecular datasets, rendering our model suitable for graphs of varying sizes.

## 2 RELATED WORK

### 2.1 DENOISING DIFFUSION MODELS FOR GRAPHS

Diffusion models (Sohl-Dickstein et al., 2015; Ho et al., 2020) have gained increasing popularity due to their impressive performance across generative tasks in computer vision (Dhariwal & Nichol, 2021; Ho et al., 2022; Poole et al., 2022), protein generation (Baek et al., 2021; Ingraham et al., 2022) or audio synthesis (Kong et al., 2020). Two core components define diffusion models. The first is a Markovian noise model, which iteratively corrupts a data point $x$ to a noisy sample $z^t$ until it conforms to a predefined prior distribution at the final step $T$. The second component is the denoising network, which is trained to revert the corrupted data to a less noisy state. This denoising network typically predicts the original data $x$ or equivalently, the added noise $\epsilon$.

After the denoising network has been trained, it can be used to sample new objects. The noise $z^T$ is firstly sampled from a prior distribution, the denoising network is then applied at each time step to predict the less noisy distribution defined by $p_\theta(z^{t-1}|z^t) = \int_x q(z^{t-1}|z^t, x) \, dp_\theta(x)$, from which the new data $z^{t-1}$ is sampled. While this integral is in general difficult to evaluate, two prominent frameworks allow for its efficient computation: Gaussian diffusion (Ho et al., 2020) and discrete diffusion (Austin et al., 2021).

When tailored to graph generation, initial diffusion models employed Gaussian noise on the adjacency matrices (Niu et al., 2020; Jo et al., 2022). They utilized a graph attention network to regress the added noise $\epsilon$. Given that $\epsilon = z^t - z$, regressing the noise is, up to an affine transformation, the same as regressing the clean graph, which is a discrete object. To keep the inherent discreteness, subsequent models (Vignac et al., 2023a; Haefeli et al., 2022) leveraged discrete diffusion and achieved top-tier results. However, such models make predictions for all pairs of nodes, which leads to a quadratic space complexity and thus restricts their scalability.

### 2.2 SCALABLE GRAPH GENERATION

Efforts to enhance the scalability of diffusion models for graph generation have mainly followed two paradigms: subgraph aggregation and hierarchical refinement.

**Subgraph Aggregation** This approach divides larger graphs into smaller subgraphs, which are subsequently combined. Notably, SnapButton (Yang et al., 2021) enhances autoregressive models

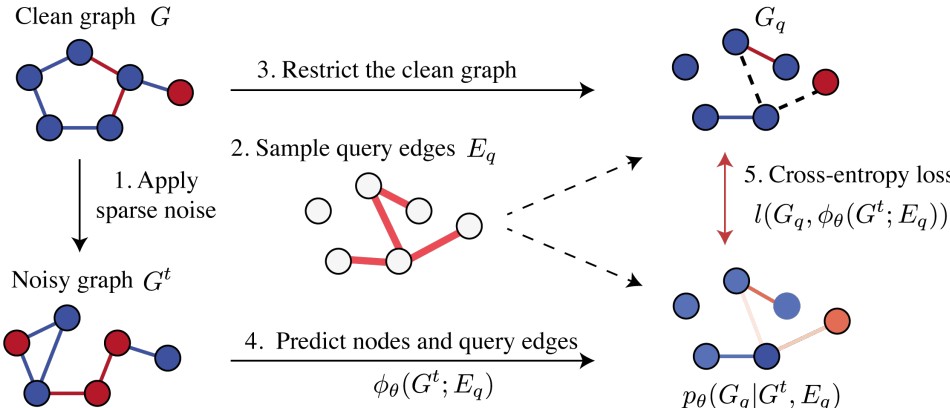

Figure 2: Overview of SparseDiff. To train a denoising neural network without considering all pairs of nodes, SparseDiff combines *i)* a noise model that preserves sparsity during diffusion; *ii)* a graph transformer $\phi_\theta$ implemented within the message-passing framework; *iii)* a loss function computed on a subset $\boldsymbol{E}_q$ of all pairs of nodes. Together, these components allow for using edge lists and training diffusion models on significantly larger graphs than dense methods.

(Liu et al., 2018; Liao et al., 2019; Mercado et al., 2021) by merging subgraphs. Meanwhile, BiGG (Dai et al., 2020) deconstructs adjacency matrices using a binary tree data structure, gradually generating edges with an autoregressive model. One notable limitation of autoregressive models is the breaking of permutation equivariance due to node ordering dependency. To counter this, (Kong et al., 2023) proposed learning the node ordering – a task theoretically at least as hard as isomorphism testing. Separately, SaGess (Limnios et al., 2023) trains a dense DiGress model to generate subgraphs sampled from a large graph and then merges these subgraphs.

**Hierarchical Refinement** This approach initially generates a low-resolution graph, which undergoes successive refinements for enhanced detail (Yang et al., 2021; Karami, 2023). For instance, the HGGT model (Jang et al., 2023) employs a hierarchical $K^2$-tree representation. Specifically for molecular generation, fragment-based models (Jin et al., 2018; 2020; Maziarz et al., 2022) adeptly assemble compounds using pre-defined molecular fragments.

A unique approach outside these paradigms was presented by EDGE (Chen et al., 2023), who initially generate a node degree distribution $\boldsymbol{d}^0$ for the nodes, and gradually craft an adjacency matrix $\boldsymbol{A}$ conditioned on this distribution along the reverse process. Despite the universal feasibility of this factorization, the ease of learning the conditional distribution $p_\theta(\boldsymbol{A}|\boldsymbol{d}^0)$ remains uncertain, as there do not always exist undirected graphs that satisfy a given degree distribution.

Overall, scalable generation models typically either introduce a dependence on node orderings or rely heavily on the existence of a community structure in the graphs. In contrast, the SparseDiff model described in the next section aims at making no assumption besides sparsity, which results in very good performance across a wide range of graphs.

## 3 SPARSEDIFF: SPARSE DENOISING DIFFUSION FOR LARGE GRAPH GENERATION

We introduce the Sparse Denoising Diffusion Model (SparseDiff), designed to bolster the scalability of discrete diffusion models for sparse graphs. Our model enables efficient training, extends the high performance of current discrete graph models to significantly larger graphs, and at the same time offers a user-friendly method for controlling GPU usage.

SparseDiff adopts the sparse graph representation, where a graph $G$ composed of $n$ nodes and $m$ edges, is denoted as a triplet $(\boldsymbol{E}, \boldsymbol{X}, \boldsymbol{Y})$. Here, $\boldsymbol{E} \in \mathbb{N}^{2 \times m}$ represents the edge list detailing indices of endpoints, while the node and edge attributes are encoded respectively with the one-hot format $\boldsymbol{X} \in \{0,1\}^{n \times a}$ and $\boldsymbol{Y} \in \{0,1\}^{m \times b}$. As illustrated in Fig. 2, our approach incorporates three core

---

**Algorithm 1:** Sparse Training at Step $t$ with predefined edge fraction $\lambda$

---

1: Sample the sparse noisy graph $G^t = (\boldsymbol{E}^t, \boldsymbol{X}^t, \boldsymbol{Y}^t)$ with the sparse noise model (3.1);
2: Sample random query edges $\boldsymbol{E}_q$ of size $\lambda n^2$ from all node pairs (3.2);
3: Construct the sparse computational graph $G_c$ whose edge list $\boldsymbol{E}_c$ contains $\boldsymbol{E}^t \cup \boldsymbol{E}_q$ (3.3.1);
4: Feed $G_c$ into the message-passing network $\phi_\theta$ and predict $\hat{\boldsymbol{X}}$ and $\hat{\boldsymbol{Y}}_q$ (3.3.2);
5: Perform sparse loss calculation on $(\boldsymbol{X}, \hat{\boldsymbol{X}})$ and $(\boldsymbol{Y}_q, \hat{\boldsymbol{Y}}_q)$ (3.2);

---

components to enable training in a space-efficient manner, and Algorithm 1 provides a step-by-step description of the training algorithm, referring to their respective explanations. Subsequent sections will elaborate on each of these three components.

### 3.1 SPARSITY-PRESERVING NOISE MODEL

To begin with, a sparse graph diffusion model involves a noise model designed to maintain sparsity throughout the diffusion process. This implies that the number of edges of the noisy graph $G^t$ must match that of $G$ to sustain consistent complexity throughout the diffusion. At the same time, the computational complexity of applying noise to a graph should remain sub-quadratic as well.

**Marginal Transition** The first requirement necessitates the adoption of a discrete diffusion model. In this framework, instead of using the noisy distribution $q(G^t|G)$ itself as the noisy data, we sample $G^t$ from $q(G^t|G)$ to keep its discrete nature. Precisely, the noisy distribution can be obtained through $q(G^t|G) = (\boldsymbol{X}\bar{\boldsymbol{Q}}_{\boldsymbol{X}}^t, \boldsymbol{Y}\bar{\boldsymbol{Q}}_{\boldsymbol{Y}}^t)$, where $\bar{\boldsymbol{Q}}^t = \boldsymbol{Q}^1\boldsymbol{Q}^2 \ldots \boldsymbol{Q}^t$ for $\boldsymbol{X}$ and $\boldsymbol{Y}$ respectively, and $\boldsymbol{Q}^t$ represents the Markov transition matrix from step $t-1$ to step $t$. While there exist different Markov transition matrices such as uniform transitions, absorbing transitions, and marginal transitions, only the last one is supported theoretically (Ingraham et al., 2022; Vignac et al., 2023a) and maintains the same level of sparsity (i.e. edge numbers) through diffusion. In the marginal transition model, the probability of transitioning to a state is proportional to the marginal probability of that state in the data. In the context of sparse graphs, this means that jumping to the state "no edge" will be very likely, as it is the dominant label in the data. Formally, if $\boldsymbol{p}_{\boldsymbol{X}}$ and $\boldsymbol{p}_{\boldsymbol{E}}$ are the marginal distribution of node and edge types and $\boldsymbol{p}'$ is the transpose of $\boldsymbol{p}$, and $\beta^t$ controls the noise intensity at step $t$ and $\alpha^t = 1 - \beta^t$, the marginal transition matrices for nodes and edges are defined by: $\boldsymbol{Q}_{\boldsymbol{X}}^t = \alpha^t\boldsymbol{I} + \beta^t\boldsymbol{1}_a\boldsymbol{p}_{\boldsymbol{X}}$ and $\boldsymbol{Q}_{\boldsymbol{Y}}^t = \alpha^t\boldsymbol{I} + \beta^t\boldsymbol{1}_b\boldsymbol{p}_{\boldsymbol{Y}}'$.

**Theoretical Analysis regarding Sparsity** We note that our choice of noise model does not guarantee that the noisy graph is always sparse. However, it is the case with high probability as stated by the following lemma, which is an application of Desolneux et al. (2008) (cf. Appendix B). This lemma shows that, in large and sparse graphs, the probability that the fraction of edges $r$ in the noisy graph is higher than the actual existing edge ratio $k$ decreases exponentially with the graph size. For instance, for $k$ small and $r = 2k$, this probability can be written with $c_1 e^{-c_2 n^2 k}$ for two constants $c_1$ and $c_2$, which is considerably small when $n$ is large.

**Lemma 3.1.** *(High-probability bound on the sparsity of the noisy graph)*
*Consider a graph with $n$ nodes and $m$ edges. If the edge ratio given by $m/(n(n-1)/2)$ is denoted as $r$, and the number of edges in the noisy graph $G^t$ sampled from the marginal transition model is given by $m_t$. Then, for $n$ sufficiently large and $k < 1/4$, for any $k < r < 1$, we have:*

$$log(\mathrm{P}[\frac{2m_t}{n(n-1)} \geq r]) \sim -\frac{n(n-1)}{2}(r log \frac{r}{k} + (1-k)log \frac{1-r}{1-k}) \tag{1}$$

**Sparse Sampling of the Noise Model** The second quadratic component arises from noise sampling. Although the marginal transition keeps the sparsity of $G^t$, to obtain the noisy graph, standard discrete diffusion models simply compute transition probabilities using a product $\boldsymbol{Y}\boldsymbol{Q}_{\boldsymbol{Y}}^t \in \mathbb{R}^{n \times n \times b}$ and sample from it. This multiplication based on the dense edge representation is however no longer compatible with our sparse edges encoded in $\boldsymbol{Y} \in \{0,1\}^{m \times b}$ and $\boldsymbol{E} \in \mathbb{N}^{2 \times m}$. To enable sparse sampling, we adopt a three-step approach to sample noisy graphs without using dense tensors.

1. We consider "existing edge" types by computing $\boldsymbol{Y}\boldsymbol{Q}_{\boldsymbol{Y}}^t \in \mathbb{R}^{m \times b}$ for edges in the edge list $\boldsymbol{E}$ of the clean graph $G$ and sample their new label from this categorical distribution;

2. We consider the "no edge" type, and determine the number of new edges to add to the list $\boldsymbol{E}^t$. This number follows a binomial distribution $\mathcal{B}(\bar{m}_t, k)$, which takes $\bar{m}_t$ draws from all edges of "no edge" type with the probability $k$ of turning into an "existing edge" state. Here, $\bar{m}_t = n(n-1)/2 - m_t$, and $k = 1 - \boldsymbol{Q}^t[0, 0]$.

3. We sample positions for these new edges uniformly from non-occupied positions in the adjacency matrix of $G$. As elaborated in Appendix A.2, an efficient and highly intricate algorithm has been devised for sampling with the sparse edge list $\boldsymbol{E}$ instead of the quadratic adjacency matrix $\boldsymbol{A}$, which ensures the sub-quadratic space complexity in computation.

## 3.2 SPARSITY-PRESERVING LOSS FUNCTION

In discrete denoising diffusion for graphs such as (Vignac et al., 2023a; Haefeli et al., 2022), a neural network is trained to predict the clean graph, i.e., the class of each node and each pair, which introduces another quadratic component to the model. To avoid this, one alternative method is to make predictions for a subset of edges instead of for all node pairs. For this purpose, SparseDiff introduces a parameter "edge fraction" $\lambda$ which corresponds to a fraction of pairs that are sampled uniformly in each forward pass. Such sampled edges are called "query edges", and denoted by $\boldsymbol{E}_q$. In our implementation, $\lambda$ was treated as a constant and chosen to balance GPU usage. The computational complexity of SparseDiff is up-bounded by $O(m + \lambda n^2)$, as the model needs to process both the noisy graph and the query edges. However, by choosing $\lambda = O(m/n^2)$, it could result in a $O(m)$ complexity as opposed to $O(n^2)$ for DiGress.

Precisely, if the constant $c$ (set to 5 in experiments) is used to balance the importance of nodes and edges, the network is trained by minimizing the cross-entropy loss between the predicted distribution and the clean graph, which is simply a sum over nodes and query edges:

$$l(\hat{p}^G, G) = \sum_{1 \leq i \leq n} \text{cross-entropy}(x_i, \hat{p}_i^X) + \frac{c}{\lambda} \sum_{(i,j) \in E_q} \text{cross-entropy}(y_{ij}, \hat{p}_{ij}^Y),$$

## 3.3 SPARSE MESSAGE-PASSING TRANSFORMER

The final component of the Sparse Diffusion Model is a memory-efficient graph neural network. In previous diffusion models for graphs, the main complexity bottleneck lay in the need to encode features for all pairs of nodes, leading to a computation complexity that scaled as $O(l\, n^2\, d_e)$, where $l$ is the number of layers and $d_e$ the dimensionality of edge activations. To address this issue, it is necessary to avoid learning embeddings for all pairs of nodes. Fortunately, as our noisy graphs are sparse, edge list representations can be leveraged. These representations can be efficiently used within message-passing neural networks (MPNNs) architectures (Scarselli et al., 2008; Gilmer et al., 2017) through the use of specialized libraries such as Pytorch Geometric (Fey & Lenssen, 2019) or the Deep Graph Library (Wang, 2019).

### 3.3.1 EDGE EMBEDDING MODULE DESIGN

The denoising network of SparseDiff has to deal with two simultaneous constraints. First, it needs to make predictions for the query edges $\boldsymbol{E}_q$. Second, in contrast to previous diffusion models, it cannot compute activations for all pairs of nodes. Although not possible within most message-passing architectures, the idea of predicting edge labels according to node features is however common in the context of *link prediction for knowledge graphs* (Zhang & Chen, 2018; Chamberlain et al., 2022; Boschin, 2023). We therefore first consider a link prediction approach to our problem.

**A first approach: graph learning as a link prediction problem** Instead of storing activations for pairs of edges, link prediction models typically only store representations for the nodes. In this framework, a graph neural network that learns embeddings for each node is coupled with an auxiliary module that predicts edges. In the simplest setting, this module simply computes the cosine similarity between node representations to predict the probability of being connected. While this approach is very memory efficient, we find that it has a slow convergence and poor overall performance in practice (cf. ablations in Appendix E.6). In particular, it fails to replicate the performance of dense denoising diffusion models, even on small graph datasets. This suggests that reconstructing the graph from node representations only, which is theoretically proved to be possible (Maehara & Rödl, 1990), might be hard to achieve in practice.

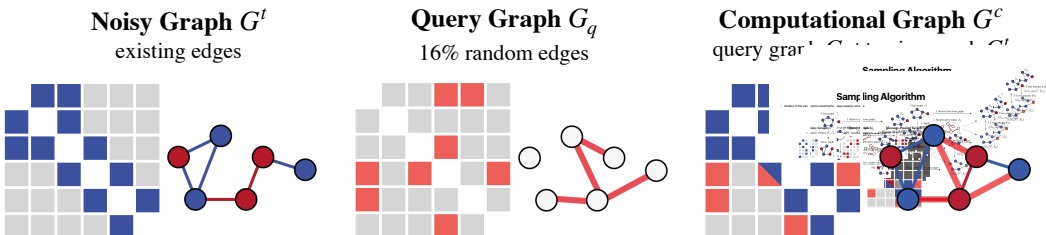

Figure 3: Definition of the noisy graph $G^t$, the query graph $G_q$, and the computational graph $G_c$, with an edge proportion $\lambda = 0.16$. The noisy graph $G^t$ is the result of our sparsity-preserving noising process, the query graph $G_q$ consists of a fraction $\lambda$ of randomly chosen edges, and the computational graph $G_c$ is the union of the noisy and query graphs. Self-loops are not included in the calculation.

**Second approach: learning representations for edges**   Based on the previous findings, we consider an approach that stores activations for pairs of nodes. The list of pairs for which we store activations will define our *computational graph* $G_c$, i.e., the graph that is used as input to the message-passing architecture. This graph contains all nodes with their noisy features $\boldsymbol{X}^t$, as well as a list of edges denoted by $\boldsymbol{E}_c$. In order to bypass the need for an edge prediction module and obtain edge features directly, the computational graph should contain the list of query edges sampled previously, i.e., $\boldsymbol{E}_q \in \boldsymbol{E}_c$. Furthermore, this graph should contain all topological information about the noisy graph, which imposes $\boldsymbol{E}^t \in \boldsymbol{E}_c$. Under these two constraints, we define the computational graph as the union of the noisy and query edge lists. Since these two graphs are sparse, the computational graph used in our message-passing architecture is guaranteed to be sparse as well.

An additional advantage of employing a computational graph that encompasses not only $G^t$ but also randomly sampled "no type" query edges is that it serves as a graph rewiring mechanism. Such edges that do not exist in the input graph $G^t$ provide the message-passing network with shortcuts, which is known to help the propagation of information and alleviate over-squashing issues (Alon & Yahav, 2020; Topping et al., 2021; Di Giovanni et al., 2023).

### 3.3.2 MODEL ARCHITECTURE

Our denoising network architecture builds upon the message-passing transformer architecture developed by Shi et al. (2020). These layers integrate the graph attention mechanism (Veličković et al., 2017) within a transformer architecture by adding normalization and feed-forward layers. In contrast to previous architectures used in denoising networks for graphs such as (Jo et al., 2022) or (Haefeli et al., 2022), the graph attention mechanism is based on edge list representations and is thus able to leverage the sparsity of graphs. We however incorporate several elements of (Vignac et al., 2023a) to improve performance. Similarly to their model, we internally manipulate graph-level features (such as the time information), as they are able to store information more compactly. Features for the nodes, edges, and graphs all depend on each other thanks to the use of PNA pooling layers (Corso et al., 2020) and FiLM layers (Perez et al., 2018) (cf. ablations in Appendix C).

Finally, we use a set of features as structural and positional encodings. These features, which include information about the graph Laplacian and cycle counts, are detailed in Appendix D. As highlighted in (Vignac et al., 2023a), these features can only be computed when the noisy graphs are sparse, which is an important benefit of discrete diffusion models. We note that not all these encodings can be computed in sub-quadratic time. However, in practice, we find that this is not an issue as these features are not back-propagated. For instance, on graphs with 500 nodes, computing these features is five times faster than the forward pass itself. Nevertheless, on even larger graphs, it might be beneficial to exclude these encodings for more efficient computation.

### 3.4 ITERATIVE SPARSE SAMPLING

Once the denoising network has been trained, it can be used to sample new graphs. Similar to other graph diffusion models, we first sample a node number $n$ and keep it constant during diffusion. Then, from the prior distribution $G^T \sim \prod_{i=1}^n \text{Cate}(\boldsymbol{p_X}) \times \prod_{1 \le i < j \le n} \text{Cate}(\boldsymbol{p_Y})$, we sample a ran-

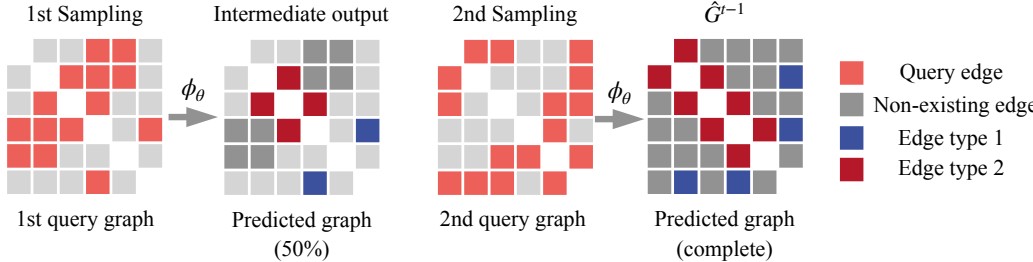

Figure 4: Visualization of the iterative sampling process, with a query edge proportion $\lambda$ of $50\%$. In the figure, SparseDiff iterates twice to cover all node pairs, with each iteration involving the sampling of new edge types for $50\%$ edges to populate the adjacency matrix.

dom graph based on the sparse sampling algorithm (c.f. 3.1), where $p_X$ and $p_Y$ are the marginal probabilities of each class in the data and $\mathrm{Cate}(p)$ denotes their corresponding categorical distribution. Note that, in the particular case of unattributed graphs, sampling from this prior distribution amounts to sampling an Erdos-Renyi graph.

After the graph $G^T$ has been sampled, the denoising network can be applied recursively. However, the full graph cannot be predicted at once, as this would first require quadratic memory, and would also create a distribution shift: as the message-passing network has been trained on sparse computational graphs $G_c^t$, dense graphs should not be used at inference time. We therefore use an iterative procedure, illustrated in Fig. 4, to cover all node pairs of $G^{t-1}$. We first consider all $n(n-1)/2$ indices representing pairs of nodes and randomly permute them. We cut the resulting array into equal-sized chunks that represent the query edge list $E_q$ at each iteration. We then iterate over these blocks, adding new edges into the edge list $E^{t-1}$, while keeping the noisy graph $G^t$ fixed. This procedure results in $\lceil \frac{1}{\lambda} \rceil$ calls to the denoising diffusion model at each diffusion step.

We note that our approach introduces higher time complexity during sampling. However, as discussed before, it is challenging to avoid quadratic predictions without assuming specific characteristics of the data distribution. Moreover, while the high space complexity makes training with traditional diffusion models almost impossible, increased sampling time does not emerge as the primary bottleneck for practice usage.

## 4 Experiments

We conduct experiments to present the capability of SparseDiff across a wide range of graphs. SparseDiff matches state-of-the-art performance on datasets of small molecules (QM9, Moses), while being simultaneously very competitive on datasets of larger graphs (Planar, SBM, Protein, Ego). We compare the performance of SparseDiff to GraphNVP (Madhawa et al., 2019), DiGress (Vignac et al., 2023a), Spectre (Martinkus et al., 2022), GraphRNN (You et al., 2018), GG-GAN (Krawczuk et al., 2017), JDSS (Jo et al., 2022), as well as several scalable models: HiGen (Karami, 2023), EDGE (Chen et al., 2023), BiGG (Dai et al., 2020) and HGGT (Jang et al., 2023), and GraphARM (Kong et al., 2023). Considering that certain datasets are conventionally assessed based on a limited number of generated samples, the variance in results for each sampling can be important. To promote a more convincing comparison, we present the average and standard deviation across 5 runs for each metric. If a result falls within the bounds of the standard deviation of our results, we also consider them comparable for fair assessment.

### 4.1 Molecule generation

Since our method considers all node pairs as dense models when $\lambda = 1$, it should match their performance on datasets of small graphs. We verify this capability on the QM9, and Moses molecular datasets used in DiGress (Vignac et al., 2023a). The QM9 dataset (Wu et al., 2018) that contains molecules with up to 9 heavy atoms can either be treated with implicit or explicit hydrogens. The Moses benchmark (Polykovskiy et al., 2020), based on ZINC Clean Leads, contains drug-sized molecules and features many tools to assess the model performance. Since QM9 contains charged

Table 1: Molecule generation on QM9 with implicit hydrogens (mean and std over 5 samplings). For a fair comparison, DiGress was modified to handle formal charges and benchmarked. While there is no major benefit to using sparsity on small graph, SparseDiff is very competitive, while other scalable models have a poor FCD metric, indicating that they do not correctly model the data.

| Class | Method | Valid (%) ↑ | Unique (%) ↑ | Connected (%) ↑ | FCD ↓ |
|-------|--------|-------------|--------------|-----------------|-------|
| Dense | SPECTRE | 87.3 | 35.7 | - | - |
| | GraphNVP | 83.1 | 99.2 | - | - |
| | GDSS | 95.7 | 98.5 | - | - |
| | DiGress | 99.2 | 95.9 | 99.5 | 0.15 |
| | DiGress + charges | $99.3_{\pm.0}$ | $95.9_{\pm.2}$ | $99.4_{\pm.2}$ | $0.15_{\pm.01}$ |
| Sparse | GraphARM | 90.3 | - | - | 1.22 |
| | EDGE | 99.1 | **100** | - | 0.46 |
| | HGGT | 99.2 | 95.7 | - | 0.40 |
| | SparseDiff(ours) | $\mathbf{99.6}_{\pm.04}$ | $99.7_{\pm.01}$ | $\mathbf{99.7}_{\pm.02}$ | $\mathbf{0.11}_{\pm.01}$ |

Table 2: Unconditional generation on the Stochastic Block Model (SBM) and Planar datasets. A SBM graph is valid if it passes a statistical test for the stochastic block model, while a planar graph is valid if it is planar and connected. Results are presented in the form of ratios: $\mathrm{MMD}(\mathrm{generated}, \mathrm{test})^2 / \mathrm{MMD}(\mathrm{train}, \mathrm{test})^2$. VUN: valid, unique & novel graphs.

| Dataset | Stochastic block model | | | | Planar | | | |
|---------|------|--------|--------|---------|------|--------|--------|---------|
| Model | Deg.↓ | Clust.↓ | Orbit↓ | V.U.N.↑ | Deg. ↓ | Clust. ↓ | Orbit↓ | V.U.N.↑ |
| GraphRNN | 6.9 | 1.7 | 3.1 | 5% | 24.5 | 9.0 | 2508 | 0% |
| GRAN | 14.1 | 1.7 | 2.1 | 25% | 3.5 | 1.4 | 1.8 | 0% |
| GG-GAN | 4.4 | 2.1 | 2.3 | 25% | – | – | – | – |
| SPECTRE | 1.9 | 1.6 | 1.6 | 53% | 2.5 | 2.5 | 2.4 | 25% |
| DiGress | **1.6** | **1.5** | 1.7 | **74**% | **1.4** | **1.2** | **1.7** | 75% |
| HiGen | 2.4 | **1.5** | **1.4** | – | – | – | – | – |
| SparseDiff | $\mathbf{2.0}_{\pm1.6}$ | $1.5_{\pm.0}$ | $\mathbf{1.4}_{\pm.1}$ | $56\%_{\pm8.5}$ | $3.6_{\pm1.7}$ | $1.4_{\pm.4}$ | $3.4_{\pm1.2}$ | $\mathbf{88\%}_{\pm7}$ |

atoms, we incorporate formal charges as an additional discrete node feature that is learned during diffusion, similarly to (Vignac et al., 2023b). For a fair comparison, we also apply this improvement to DiGress.

For the QM9 dataset, we assess performance by checking the proportion of connected graphs, the molecular validity of the largest connected component (measured by the success of RDKit sanitization), and the uniqueness of over 10,000 molecules. Additionally, we use the Frechet ChemNet Distance (FCD) (Preuer et al., 2018) which measures the similarity between sets of molecules using a pretrained neural network.

In Table 1, we observe that SparseDiff overall achieves the best performance on QM9 with implicit hydrogens except on uniqueness. In particular, it clearly outperforms other scalable methods on the FCD metric, showing that such methods are not well suited to small and very structured graphs. Results for QM9 with explicit hydrogens and the MOSES dataset are presented in Tables 6 (Appendix E.3), and Table 7 (Appendix E.4). We find that SparseDiff compares similarly to the DiGress model. This result confirms further our performance on small and not highly sparse datasets.

## 4.2 LARGE GRAPH GENERATION

We also evaluate our model on datasets of graphs with increasing size. First, we test our model's ability to generate graphs without edge crossings through a dataset of planar graphs (with 64 nodes per graph). Then, we consider a dataset drawn from the Stochastic Block Model (SBM) (Martinkus et al., 2022) with 2 to 5 communities. Its graphs contain up to 200 nodes, which is the largest size used in dense diffusion models such as DiGress (Vignac et al., 2023a). Finally, we use the Ego (Sen et al., 2008) and Protein (Dobson & Doig, 2003) datasets that feature graphs with up to 500 nodes. Ego is sourced from the CiteSeer (Giles et al., 1998) dataset and captures citation relationships,

Table 3: Unconditional generation on graphs with up to 500 nodes. On such graphs, dense models such as DiGress clearly fail, whereas SparseDiff presents competitive performance on most metrics. Results are presented in the form of ratios: $\mathrm{MMD}(\text{generated}, \text{test})^2 / \mathrm{MMD}(\text{train}, \text{test})^2$.

| Dataset | Class | Model | Degree ↓ | Clustering ↓ | Orbit ↓ | Spectre ↓ | RBF ↓ |
|---------|-------|-------|----------|--------------|---------|-----------|-------|
| *Protein* | Dense | GRAN | 6.7 | 7.1 | 40.6 | 5.7 | – |
| | | DiGress | $18.4_{\pm 2.3}$ | $14.8_{\pm 2.1}$ | $16.0_{\pm 5.5}$ | $5.9_{\pm 1.1}$ | 5.2 |
| | Sparse | DRuM | 6.3 | 9.7 | 10.8 | 3.3 | – |
| | | BiGG | **0.3** | 3.7 | **7.1** | 5.0 | – |
| | | HiGen | 4.0 | 6.4 | 7.3 | 2.8 | – |
| | | SparseDiff | $10.3_{\pm 1.1}$ | $\mathbf{3.4}_{\pm .2}$ | $15.1_{\pm 2.0}$ | $\mathbf{1.6}_{\pm .07}$ | $\mathbf{3.4}_{\pm .7}$ |
| *Ego* | Dense | DiGress | 354 | 0.9 | 100 | – | 5.3 |
| | Sparse | EDGE | 290 | 17.3 | 4.3 | – | 4.0 |
| | | HiGen | 236 | **0.3** | 3.2 | – | **3.7** |
| | | SparseDiff | $\mathbf{9.5}_{\pm 3.5}$ | $5.4_{\pm .2}$ | $\mathbf{2.5}_{\pm .1}$ | $3.6_{\pm 1.1}$ | $3.9_{\pm 1.0}$ |

Table 4: Convergence comparison after being trained for different time with Ego dataset.

| Training time | 2 days | | | | 4 days | | | |
|---------------|--------|--------|---------|--------|--------|--------|---------|--------|
| Metrics | Deg.↓ | Orbit↓ | Clust.↓ | Spec.↓ | Deg.↓ | Orbit↓ | Clust.↓ | Spec.↓ |
| DiGress | 0.042 | 0.185 | 0.208 | 0.013 | 0.033 | 0.144 | 0.216 | 0.011 |
| SparseDiff | **0.004** | **0.053** | **0.069** | **0.007** | **0.002** | **0.036** | **0.059** | **0.004** |

while Protein represents amino acids connected when they are within 6 Angstroms of each other. Statistics for these datasets can be found in Appendix E.2.

For evaluation, we first include MMD metrics, which are commonly used in graph generation tasks. As MMD metrics usually produce small values that are challenging to compare directly, we report metrics divided by $\mathrm{MMD}(\text{training}, \text{test})^2$, and provide our raw results in Appendix E.5. We also use the RBF MMD metric defined in Thompson et al. (2022) to measure the diversity and fidelity of generated graphs using a randomly parametrized GNN. Besides, we especially report the validity of generated graphs for the SBM dataset, which is the fraction of graphs that pass a statistical test for the stochastic block model. For the Planar dataset, validity corresponds to the fraction of graphs that are planar and connected.

Results are presented in Tables 2 and 4. Although dense models demonstrate excellent performance on mid-sized graphs like SBM and planar graphs, they struggle with larger graphs. This is due to the necessity of using a small batch size (e.g., 2 on a 32GB GPU) for such graphs, resulting in slow training and poor convergence. In contrast, SparseDiff not only matches previous dense and sparse models on mid-sized datasets but also remains competitive with scalable models across various metrics on large datasets.

**Efficiency Analysis** To showcase the empirical efficiency improvement during training of SparseDiff over its limit case DiGress, Table 4 illustrates the significantly faster convergence of SparseDiff on the Ego dataset. More precisely, SparseDiff achieved superior results compared to a DiGress model trained for 4 days, even after only 2 days of training.

## 5 CONCLUSION

In this study, we introduce SparseDiff, a scalable discrete denoising diffusion model for graph generation. SparseDiff provides high controllability over GPU usage and permits the use of edge list representations by predicting only a subset of edges at once. Experimental results demonstrate that SparseDiff exhibits high performance across all graph sizes, whereas other scalable methods tend to perform poorly on small, structured graphs. SparseDiff enhances the capabilities of discrete diffusion models to process larger datasets, thereby broadening its applicability, including tasks such as generating large biological molecules and community graphs, among others.

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
