## A    ALGORITHM

### A.1    GENERATION ALGORITHM

In the process of graph generation, we employ an iterative approach to construct the adjacency matrix. During each iteration, a random set of query edges, counting a proportion denoted by $\lambda$, is drawn without repetition from all edges until the entire matrix is populated. It is worth noting that when $\lambda$ does not evenly divide 1, the last iteration may result in a different number of query edges. To maintain consistency in the number of query edges, we adopt a strategy in such cases: we utilize the last $\lambda$ percent of edges from the dataset. This suggests that a small portion of edges will be repeatedly sampled during generation, but given that their predicted distribution should remain the same, this strategy will not change any mathematical formulation behind it. Besides, the nodes are only sampled once across all iterations. To enhance clarity and facilitate understanding, we provide Algorithm 2 as the following.

---

**Algorithm 2:** Sampling from SparseDiff

---

Sample the number of nodes $n$ from the training data distribution
Sample $G^T \sim q_{\boldsymbol{X}}(n) \times q_{\boldsymbol{Y}}(n)$                                  *// Random graph*
**for** $t = T$ **to** $1$ **do**

    Permute $\boldsymbol{E}(n)$                                         *// Permuted list of all pairs between $n$ nodes*
    Partition $\boldsymbol{E}(n)$ to $\boldsymbol{E}_p = [\boldsymbol{E}_1, ..., \boldsymbol{E}_{1/\lambda}]$                    *// $\boldsymbol{E}_i$ has equal length*
    Create the empty graph $G^{t-1}$
    **for** $i = 1$ **to** $1/\lambda$ **do**

        Obtain the query graph $G_q$ from $\boldsymbol{E}_p$                    *// $G_q$ contains edges in $\boldsymbol{E}_p[i]$*
        $z \leftarrow f(G^t, t)$                            *// Optional: structural and spectral encoding*
        $\hat{p}^{\boldsymbol{X}}, \hat{p}_q^{\boldsymbol{Y}} \leftarrow \phi_\theta(G^t, G_q, z)$                            *// Sparse forward pass*
        Sample query edge labels $\boldsymbol{Y}^q$ according to $\hat{p}_q^{\boldsymbol{Y}}$
        Add $\boldsymbol{Y}_{new}^q$ and $\boldsymbol{E}_{new}^q$ to $G^{t-1}$                    *// Only add "existing type" edges*
        **if** $i == 0$ **then**

            Sample node labels $\boldsymbol{X}$ according to $\hat{p}^{\boldsymbol{X}}$
            Update $\boldsymbol{X}$ to $G^{t-1}$                            *// Update nodes only once*
        **end**

    **end**

**end**
**return** $G^0$

---

### A.2    SPARSE NOISE MODEL

We design a special algorithm to apply noise to graph data in a sparse manner. The fundamental idea behind this algorithm is to treat separately "existing" type and "no edge" (i.e. non-existing) type edges. In sparse graphs, the number of edges typically scales sub-quadratically with the number of nodes, denoted as $n$, while the quadratic space complexity mainly stems from the "no edge" type edges. As a step-by-step description has been provided in 3.1, the most challenging aspect of this algorithm is to randomly attribute new edges to the set of non-occupied node pairs without introducing the quadratic adjacency matrix. Given its seeming simplicity, the challenge is mainly due to the need to: i) avoid loops for computational efficiency, ii) respect the batching mechanism of PyTorch geometric, which does not use an extra batch size dimension in the tensors, and iii) handle graphs of varying sizes. Due to the algorithm's complexity and the technical intricacies involved, a more detailed description of our algorithm is discussed later in Appendix A.3, and the technical details are provided in our code base.

### A.3    UNIFORM SAMPLING FOR NON-EXISTING EDGES

When sampling non-existing edges, a common approach is to use the adjacency matrix, which can be problematic for large graphs due to its quadratic size. The same challenge arises in the final step of sampling sparse noise.

Consider a graph with 5 nodes, featuring 4 existing edges and 6 pairs of nodes that are not connected. The condensed indices of the existing edges are 0, 3, 4, and 6. If the objective is to sample 2 non-existing edges, you can start by randomly selecting two indices from the range $[0, ..., 5]$, which corresponds to the 6 non-existing edges. For example, if indices $[2, 3]$ are randomly chosen, where 2 denotes the position of the third non-existing edge, and 3 represents the fourth non-existing edge. These condensed indices are then inserted in the list of non-existing edges. Upon amalgamation with the existing edges, the final set of edges will become $[5, 7]$. This approach allows us to efficiently sample non-existing edges while ensuring the proper placement of existing edges within the sampled set. Please refer to our codes for more implementation details.

## B  PROOF OF LEMMA 3.1

The lemma for a noisy graph with guaranteed sparsity comes directly from the proposition regarding the tail behavior of a binomial distribution (Desolneux et al., 2008) as follows:

**Proposition B.1.** *(Tail behavior of a binomial distribution)*

*Let $X_i, i = 1, ...l$ be independent Bernoulli random variables with parameter $0 < p < \frac{1}{4}$ and let $S_l = \sum_{i=1}^{l} X_i$. Consider a constant $p < r < 1$ or a real function $p < r(l) < 1$. Then according to the Hoeffding inequality, $\mathcal{B}(l, k, p) = \mathbb{P}[S_l \geq k]$ satisfies:*

$$-\frac{1}{l} log \mathcal{P}[S_l \geq rl] \geq r log \frac{r}{p} + (1-p) log \frac{1-r}{1-p} \quad (2)$$

For sparse graphs, the edge ratio $k$ is clearly smaller than $\frac{1}{4}$. Consider then Bernoulli random variables with parameter $k$ and a constant $k < r < 1$ with $n(n-1)/2$ (i.e. number of all node pairs in an undirected graph) draws, and note sampled existing edge number $S_{n(n-1)/2}$ as $m_t$, we have:

$$log(\mathbb{P}[\frac{m_t}{n(n-1)/2} \geq r]) \leq -\frac{n(n-1)}{2}[r log \frac{r}{k} + (1-k) log \frac{1-r}{1-k}] \quad (3)$$

## C  MODEL ARCHITECTURE

We introduce the FiLM layer and the PNA layer inside the model architecture to enhance its performance. Precisely, the FiLM layer is used to combine features at different scales, such as node and edge features. Specifically, given two features $M_1$ and $M_2$, and trainable parameters $W_1$ and $W_2$, the FiLM layer output is calculated as $\text{FiLM}(M_1, M_2) = M_1 W_1 + (M_1 W_2) \odot M_2 + M_2$. As an illustration, within the convolutional layer, the graph feature $M_2$ is integrated with edge features $M_1$ to enhance predictions. While PNA layer is used as a specialized pooling layer to obtain information from different dimensions of a specific feature. Given the feature X and trainable parameter $W$, $\text{PNA}(X) = \text{cat}(\max(X), \min(X), \text{mean}(X), \text{std}(X)) W$. For example, node features $X$ are forwarded to a PNA layer for extracting global information across different scales, which is subsequently added to the graph feature to enhance its representation.

## D  STRUCTURAL AND POSITIONAL ENCODINGS

During training, we augment model expressiveness with additional encodings. To make things clear, we divide them into encodings for edges, nodes, and for graphs.

**Encoding for graphs**  We first incorporate graph eigenvalues, known for their critical structural insights, and cycle counts, addressing message-passing neural networks' inability to detect cycles (Chen et al., 2020). The first requires $n^3$ operations for matrix decomposition, the second $n^2$ for matrix multiplication, but both are optional in our model and do not significantly limit scalability even with graphs up to size 500. In addition to the previously mentioned structural encodings, we integrate the degree distribution to enhance the positional information within the graph input, which is particularly advantageous for graphs with central nodes or multiple communities. Furthermore, for

graphs featuring attributed nodes and edges, the inclusion of node type and edge type distributions also provides valuable benefits.

**Encoding for nodes**   At the node level, we utilize graph eigenvectors, which are fundamental in graph theory, offering valuable insights into centrality, connectivity, and diverse graph properties.

**Encoding for edges**   To aid in edge label prediction, we introduce auxiliary structural encodings related to edges. These include the shortest distance between nodes and the Adamic-Adar index. The former enhances node interactions, while the latter focuses on local neighborhood information. Due to computational constraints, we consider information within a 10-hop radius, categorizing it as local positional information.

**Molecular information**   In molecular datasets, we augment node features by incorporating edge valency and atom weights. Additionally, formal charge information is included as an additional node label for diffusion and denoising during training, as formal charges have been experimentally validated as valuable information (Vignac et al., 2023b).

## E   ADDITIONAL EXPERIMENTS

### E.1   MMD METRICS

In our research, we carefully select specific metrics tailored to each dataset, with a primary focus on four widely recognized Maximum Mean Discrepancy (MMD) metrics. These metrics utilize the total variation (TV) distance, as detailed in (Martinkus et al., 2022). They encompass node degree (Deg), clustering coefficient (Clus), orbit count (Orb), and graph spectra (Spec). The first three local metrics compare the degree distributions, clustering coefficient distributions, and the occurrence of all 4-node orbits within graphs between the generated and training samples. Additionally, we extend our analysis to include the comparison of graph spectra by computing the eigenvalues of the normalized graph Laplacian, providing complementary insights into the global properties of the graphs.

### E.2   STATISTICS OF THE DATASETS

To provide a more comprehensive overview of the various scales found in existing graph datasets, we present here key statistics for them. These statistics encompass the number of graphs, the range of node numbers, the range of edge numbers, the edge fraction for existing edges, and the query edge proportion $\lambda$ used for training, i.e. the proportion of existing edges among all node pairs. In our consideration, we focus on undirected graphs. Therefore, when counting edges between nodes $i$ and $j$, we include the edge in both directions.

Table 5: Statistics for the datasets employed in our experiments.

| Name | Graph number | Node range | Edge range | Edge Fraction (%) | $\lambda$ (%) |
|------|-------------|-----------|-----------|------------------|--------------|
| QM9 | 133,885 | [2,9] | [2, 28] | [20, 56] | 50 |
| QM9(H) | 133,885 | [3, 29] | [4, 56] | [7.7, 44] | 50 |
| Moses | 1,936,962 | [8, 27] | [14, 62] | [8.0, 22] | 50 |
| Planar | 200 | [64, 64] | [346, 362] | [8.4, 8.8] | 50 |
| SBM | 200 | [44, 187] | [258, 2258] | [6.0, 17] | 25 |
| Ego | 757 | [50, 399] | [112, 2124] | [1.2, 11] | 10 |
| Protein | 918 | [100, 500] | [372, 3150] | [8.9, 6.7] | 10 |

### E.3   QM9 WITH EXPLICIT HYDROGENS

We additionally report the results for QM9 with explicit hydrogens in Table 6. Having explicit hydrogens makes the problem more complex because the resulting graphs are larger. We observe that SparseDiff achieves better validity than DiGress and has comparable results on other metrics when both are utilizing charges.

Table 6: Unconditional generation on QM9 with explicit hydrogens. On small graphs such as QM9, sparse models are not beneficial, but SparseDiff still achieves very good performance.

| Model | Connected | Valid↑ | Unique↑ | Atom stable↑ | Mol stable↑ |
|---|---|---|---|---|---|
| DiGress | − | 95.4 | 97.6 | 98.1 | 79.8 |
| DiGress + charges | 98.6 | 97.7 | 96.9 | 99.8 | 97.0 |
| SparseDiff | 98.1 | 97.9 | 96.9 | 99.7 | 95.7 |

Table 7: Mean and standard deviation across 5 samplings on the MOSES benchmark. SparseDiff has a similar performance to DiGress, despite a shorter training time.

| Model | Connected ↑ | Valid (%) ↑ | Unique (%) ↑ | Novel (%) ↑ | Filters (%) ↑ |
|---|---|---|---|---|---|
| GraphINVENT | − | **96.4** | 99.8 | − | 95.0 |
| DiGress | − | 85.7 | **100.0** | 95.0 | **97.1** |
| SparseDiff | $98.2_{\pm.0}$ | $86.7_{\pm.2}$ | $100.0_{\pm.0}$ | $96.3_{\pm.1}$ | $96.7_{\pm.1}$ |

| Model | FCD ↓ | SNN (%) ↑ | Scaf (%) ↑ | Frag (%) ↑ | IntDiv (%) ↑ |
|---|---|---|---|---|---|
| GraphINVENT | 1.22 | **53.9** | 12.7 | 98.6 | **85.7** |
| DiGress | **1.19** | 52.2 | **14.8** | 99.6 | 85.3 |
| SparseDiff | $1.35_{\pm.02}$ | $51.0_{\pm.0}$ | $14.2_{\pm1.7}$ | $99.6_{\pm.0}$ | $85.5_{\pm.0}$ |

| Model | Filters (%) ↑ | logP ($e^{-2}$) ↓ | SA ↓ | QED ($e^{-3}$) ↓ | Weight (%) ↓ |
|---|---|---|---|---|---|
| GraphINVENT | 95.0 | **0.67** | 4.5 | **0.25** | 16.1 |
| DiGress | **97.1** | 3.4 | **3.6** | 2.91 | 1.42 |
| SparseDiff | $96.7_{\pm.1}$ | $8.0_{\pm.4}$ | $7.9_{\pm.3}$ | $4.17_{\pm.33}$ | $1.25_{\pm.1}$ |

## E.4 MOSES BENCHMARK EVALUATION

Moses is an extensive molecular dataset with larger molecular graphs than QM9, offering a much more comprehensive set of metrics. While autoregressive models such as GraphINVENT are recognized for achieving higher validity on this dataset, both SparseDiff and DiGress exhibit advantages across most other metrics. Notably, SparseDiff closely aligns with the results achieved by DiGress, affirming the robustness of our method on complex datasets.

## E.5 RAW RESULTS

To ease comparison with other methods, Table 8 provides the raw numbers (not presented as ratios) for the SBM, Planar, Ego, and Protein datasets. Not that this table contains the FID metrics from (Thompson et al., 2022), which we did not include in the main text. The reason is that we found this metric to be very brittle, with some evaluations giving a very large value that would dominate the mean results.

Table 8: Raw results on the SBM, Planar, Protein, and Ego datasets.

| Model | Deg (e-3)↓ | Clus (e-2)↓ | Orb (e-2)↓ | Spec (e-3)↓ | FID↓ | RBF MMD (e-2)↓ |
|---|---|---|---|---|---|---|
| *SBM* | | | | | | |
| Training set | 0.8 | 3.32 | 2.55 | 5.2 | 16.83 | 3.13 |
| SparseDiff | $1.6_{\pm1.3}$ | $4.97_{\pm0.04}$ | $3.46_{\pm0.04}$ | $4.3_{\pm0.7}$ | $5.71_{\pm7.01}$ | $5.04_{\pm0.21}$ |
| *Planar* | | | | | | |
| Training set | 0.2 | 3.10 | 0.05 | 6.3 | 0.19 | 3.20 |
| SparseDiff | $0.7_{\pm0.4}$ | $4.47_{\pm1.38}$ | $0.17_{\pm0.06}$ | $6.8_{\pm0.8}$ | $4.51_{\pm1.76}$ | $5.27_{\pm0.50}$ |
| *Protein* | | | | | | |
| Training set | 0.3 | 0.68 | 0.32 | 0.9 | 5.74 | 0.68 |
| SparseDiff | $3.1_{\pm0.0}$ | $2.28_{\pm0.10}$ | $4.82_{\pm.64}$ | $1.4_{\pm.1}$ | $4.83_{\pm1.48}$ | $2.29_{\pm.48}$ |
| *Ego* | | | | | | |
| Training set | 0.2 | 1.0 | 1.20 | 1.4 | 1.21 | 1.23 |
| SparseDiff | $1.9_{\pm0.7}$ | $5.37_{\pm0.24}$ | $2.99_{\pm.17}$ | $5.0_{\pm1.5}$ | $16.15_{\pm12.86}$ | $4.83_{\pm1.18}$ |

Table 9: Influence of including edges features for edge prediction.

| Model | Deg ↓ | Clus ↓ | Orb↓ | Spec↓ | FID↓ | RBF MMD↓ |
|---|---|---|---|---|---|---|
| Link Pred | 0.0043 | 0.0721 | **0.0275** | 0.0344 | 1.51e6 | **0.0315** |
| SparseDiff | **0.0019**$_{\pm.00}$ | **0.0537**$_{\pm.00}$ | 0.0299$_{\pm.00}$ | **0.0050**$_{\pm.00}$ | **16.1**$_{\pm12.9}$ | 0.0483$_{\pm.01}$ |

Table 10: Influence of edge loss distribution on EGO dataset.

| Loss based on | Deg ↓ | Clus ↓ | Orb↓ | Spec↓ | FID↓ | RBF MMD↓ |
|---|---|---|---|---|---|---|
| Comp graph | 0.0021 | 0.0566 | **0.0270** | 0.0100 | 28.2 | **0.0396** |
| Query graph | **0.0019**$_{\pm.00}$ | **0.0537**$_{\pm.00}$ | 0.0299$_{\pm.00}$ | **0.0050**$_{\pm.00}$ | **16.1**$_{\pm12.9}$ | 0.0483$_{\pm.01}$ |

### E.6 ABLATIONS

This part presents 2 ablation experiments that motivate our approach. SparseDiff builds upon an experimental observation and a hypothesis. Firstly, our experiments demonstrate that relying solely on node features for link prediction yields unsatisfactory results. This observation encouraged us to design the computational graph that contains all edges to be predicted (i.e. query edges) as the input graph. Secondly, we hypothesized that preserving the same distribution of edge types as observed in dense graphs for loss calculation is advantageous for training. This hypothesis requires only calculating losses on uniformly sampled query edges.

#### E.6.1 LINK PREDICTION

In this experiment, we intentionally avoided using easily learnable molecular datasets that come with rich supplementary encodings. Instead, we chose to conduct the experiments on a large dataset, namely Ego, to assess their performance. In Table 9, a model that does not specifically include edge features for edge prediction performs much worse on all metrics except on RBF MMD and orbit. This observation shows that, despite that a model can also leverage the information of existing edges into node features, the lack of non-existing edges participating directly in training still ruins its performance.

#### E.6.2 QUERY EDGES WITH PROPER DISTRIBUTION

In order to emphasize the importance of preserving the edge distribution when computing losses, we conduct an experiment where we assess the performance of a model trained using all computational edges as opposed to solely using query edges. The former results in an increased emphasis on existing edges during training compared to SparseDiff. Similarly, we use the Ego dataset for initial experiments. Table 10 shows that using edges of the computational graph $G_c$ results in worse performance on most of the metrics, which indicates the importance of keeping a balanced edge distribution for loss calculation.

## F VISUALIZATION

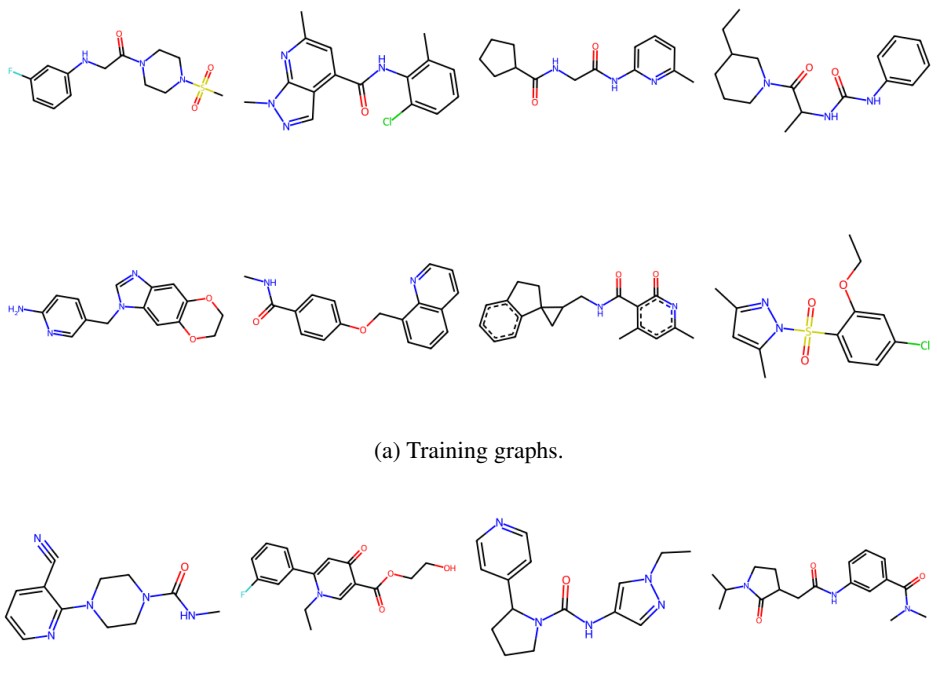

(a) Training graphs.

(b) Generated graphs.

Figure 5: Visualization for Moses dataset.

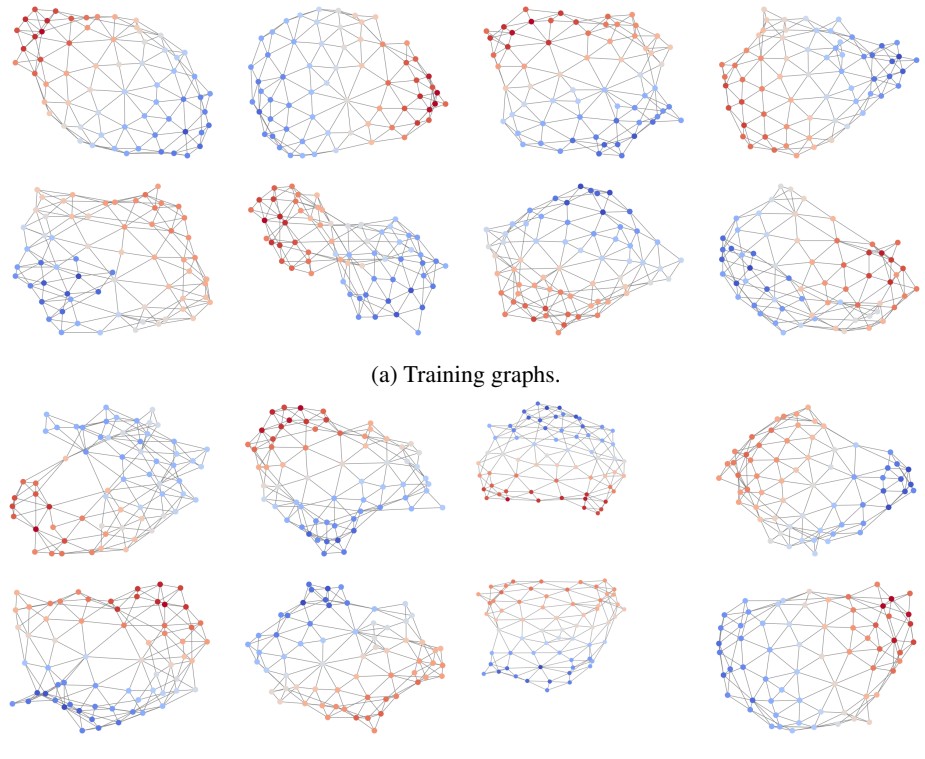

(a) Training graphs.

(b) Generated graphs.

Figure 6: Visualization for Planar dataset.

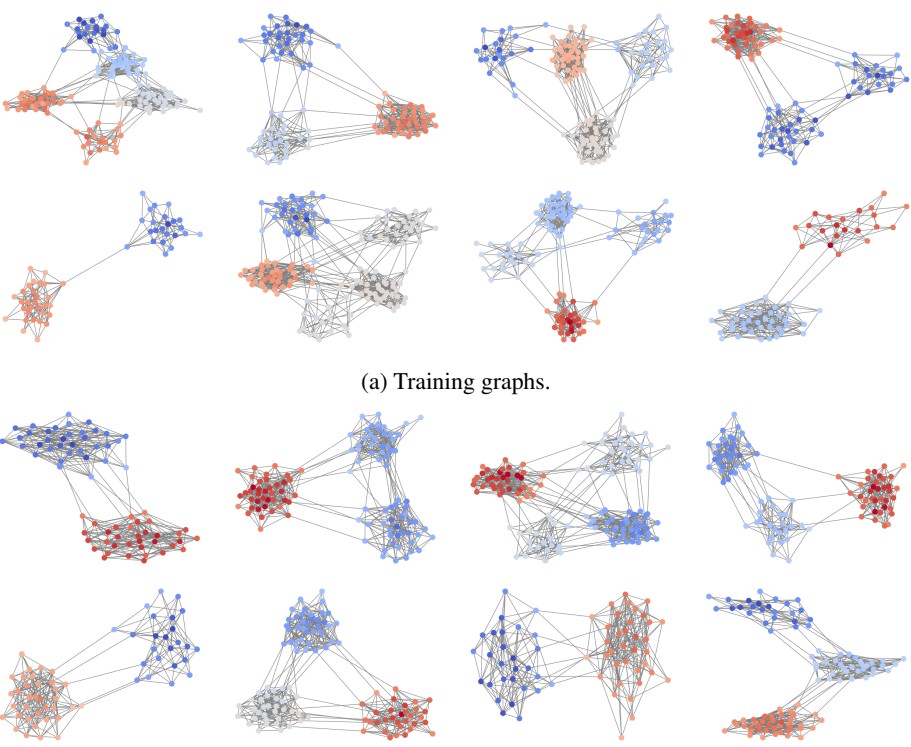

(a) Training graphs.

(b) Generated graphs.

Figure 7: Visualization for SBM dataset.

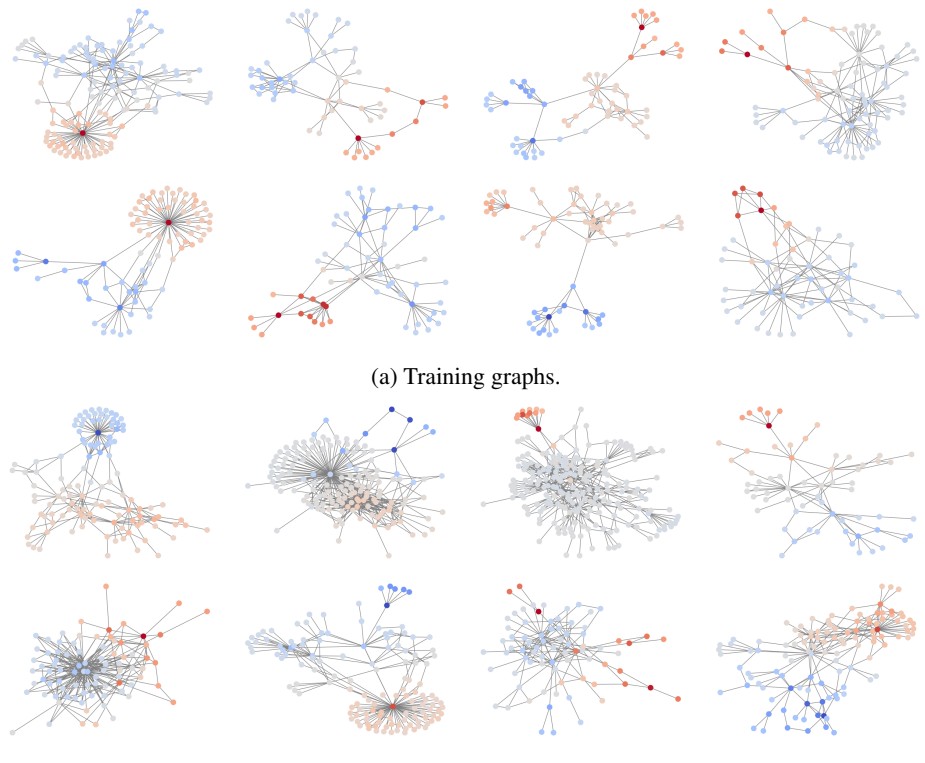

(a) Training graphs.

(b) Generated graphs.

Figure 8: Visualization for Ego dataset.

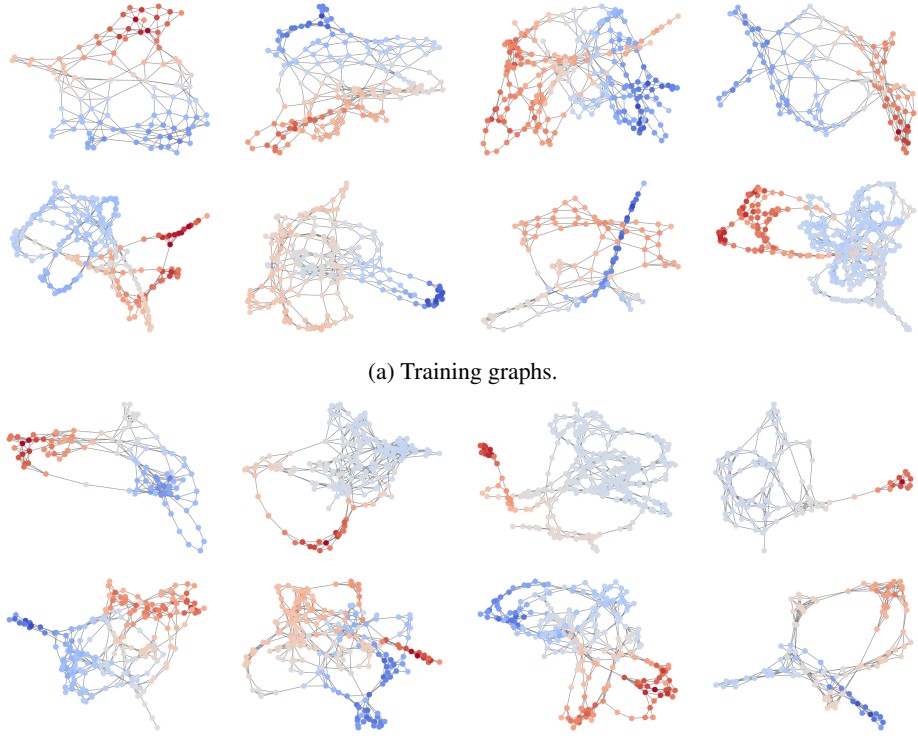

(a) Training graphs.

(b) Generated graphs.

Figure 9: Visualization for Protein dataset.