# OpenReview forum: "Sparse Training of Discrete Diffusion Models for Graph Generation"
_ICLR.cc/2024/Conference — Submitted to ICLR 2024_

### Official Review · Reviewer_teFB · 2023-10-29

**Soundness:** 3 good
**Presentation:** 3 good
**Contribution:** 2 fair
**Rating:** 5
**Confidence:** 3

**Summary:**

This work proposes SparseDiff, a diffusion-based deep method for graph generation. SparseDiff uses a message-passing graph transformer and random sampling of node pairs to compute its loss function, which allows it to exploit sparsity during the training phase. During sampling, the network repeatedly makes predictions on different subsets of node pairs until all possible edges are covered. Experiments on both small (molecular) datasets and large datasets (up to 500 node community graphs) indicate that the quality of graphs generated by SparseDiff is competitive with prior methods in both regimes.

**Strengths:**

- The paper's area of deep models for graph generation, especially diffusion models, has seen a lot of interest recently.
- The text is generally grammatical and clear.
- The core concept of sampling a fraction of node pairs to include in message-passing seems reasonable.

**Weaknesses:**

- The formatting, specifically the margins, may violate the guidelines. (Minor: citations also do not link properly when clicked.)
- There could be some discussion of exchangeability, which is concept related to the sampling of sparse graphs.
- Some notation is unclear, e.g., in the main equation on page 4, $\alpha^t$ and $\beta^t$ are not defined.
- The method is described in pieces across the text, decreasing clarity in my opinion; a step-by-step description of the algorithm is deferred to the appendix.
- There is no theoretical guarantee/characterization of the generative model's capability.
- If I have not misunderstood something, the method can only create discrete node/edge features. Also, as the authors note, the proposed method's sampling complexity scales quadratically in the number of nodes, so it does not scale to very large graphs. However, as they also note, this may be necessary without assumptions about the data distribution.

Typos
- page 4: quotation marks around "no edge"
- page 4: "markovian" uncapitalized
- page 9: "SparseDiff comparable"

**Questions:**

- Is it true that the proposed method cannot create continuous (non-categorical) node/edge features?
- "In this work, we choose to use the marginal transitions as they are supported by theoretical analysis." Could you expand on this?
- On page 7, it is noted that the method is designed to avoid distribution shifts involving predictions on the computational graphs. One vague question this raised for me is why it is then possible to exploit sparsity only during the training phase. Is the network during training not being trained to predict "non-edge" for the appropriate node pairs? Does this also constitute a distribution shift?
- To clarify what exactly the method is, I would recommend breaking out the training and sampling procedures into algorithms outside the text. There seems to be an algorithm given for the sampling procedure in the appendix, which could maybe be moved to the main paper with the space saved from fixing the margins. Ideally that algorithm could be written in a more self-contained way

---

> ### Author Response · Authors · 2023-11-16
> **Answer to Reviewer teFB**
>
> Dear reviewer,
>
> Thank you for your thorough review of our paper and the valuable feedback provided. We highly appreciate your theoretical insights and have diligently addressed the mentioned typo errors.
>
> To clarify the weaknesses mentioned in the reviews, we provide detailed responses to your comments below:
> 1. Regarding format issues: thank you for noticing the issues with citations, which have now been fixed. We are surprised by the mention of margin violations, as we followed the official latex package for ICLR2024.
> 2. Exchangeability: we thank the reviewer for this suggestion. Despite introducing some randomness in the model, SparseDiff is still an exchangeable method. The reason is that whenever we need to sample pairs, we do it uniformly on the set of all pairs so that it does not depend on one particular permutation. We will add a detailed proof in the Appendix. We are currently running experiments where we deliberately break exchangeability (as in SwinGNN) by introducing absolute positional encodings of the nodes. We will amend our current response and add the results in the Appendix when the experiments are complete, but it currently seems that breaking exchangeability tends to lead to overfitting on datasets with few graphs.
> 3. Definition clarification: Thank you for pointing this out. We have added a definition for the noise schedule.
> 4. Writing logic in the Method section: Thank you for your remark about the structure of the Method part. We took them into account in the revision of the manuscript, added the training algorithm, and improved the structure of this section.
> 5. Theoretical guarantee of the model: Establishing a solid theoretical foundation for discrete diffusion models on graphs is a challenging task. In addition to the challenges that are common to all denoising diffusion models, there are some challenges that are specific to graph generation. The biggest one is that graph neural networks (including graph Transformers) are not universal approximators of equivariant functions on graphs. As a result, even with infinite data, memory, and compute, it is not clear that standard denoising diffusion models for graphs can always recover the true data distribution. This is the case for SparseDiff as well, which explains why we only provide a limited theoretical analysis. We however believe that the experimental results are quite convincing and demonstrate the superior performance of SparseDiff.
>
> We provide responses to your additional questions:
>
> **Continuous edge and node features**
> We only treat discrete node features for the sake of simplicity. Continuous node features can be generated as well, except that they would use Gaussian noise rather than discrete diffusion.
> However, continuous edge features would be more challenging to handle, as they would require a continuous noise model that still maintains sparsity. This might be possible with truncated Gaussian distributions, but it is not a trivial extension.
>
>
>
>
> **Theoretical support for marginal transitions** Two papers support the use of marginal transitions. In essence, they both build upon the idea that generation should be easier when the distribution of noisy graphs is close to the distribution of real graphs.
> * Ingraham et al. 22, Appendix C.1: this section proves that, if $f(x) = x^T A x$ is a quadratic form that represents a property of data point x, then the expectation of $f(x_t)$ will be an interpolation of the expectation of $f(x_0)$ and $f(x_T)$. In the case of graphs, it implies that when marginal transitions are used, the noisy graphs at each time step have the same marginal distribution of node and edge types as the data.
> * Vignac et al. 23, Theorem 4.1: DiGress indicates that marginal transitions can be seen as the product distribution that minimizes the l2 distance to the true data distribution.
>
> **Concerning distribution shift** Thank you for pointing out this important aspect. We were careful to avoid distribution shifts in the design of SparseDiff:
> - During training, the model randomly samples query edges from all node pairs and calculates loss for these query edges. As shown in Figure 2, query edges are uniformly sampled from all possible node pairs. Therefore, query edges can include both existing and non-existing edges.
> - During sampling, query edges are sampled uniformly among all pairs as well. Some of these pairs will correspond to the noisy graphs, and some other pairs to the non-existing edges. The number of pairs that are sampled is the same as during training.
>
> In both cases, the graph used for message-passing is the union of the query and noisy graphs, and this graph therefore has similar statistics during training and sampling.
>
> **Clarity of the paper** Thank you very much for your suggestion. We do our best to include the algorithms in the main text within the page limit.
>
> We appreciate your review of our work and look forward to better meeting the review requirements through revisions.

---

> > ### Comment · Reviewer_teFB · 2023-11-21
> >
> > Thank you for the thorough response.
> >
> > >We are surprised by the mention of margin violations
> >
> > Perhaps I am mistaken, but I recall the margins being much larger prior to this revision, limiting the amount of content. In any case, it seems fine now.
> >
> > Thank you also for the discussion of exchangeability, marginal transitions, and distribution shift. The response on each item has addressed my concerns, and I have raised my score a bit.
> >
> > Thank you also for the discussion of the difficulty of providing theoretical guarantees. The latter seems to be a problem for papers in this subfield in general, and it seems like papers may be chasing a few (possibly limited) metrics rather than something more solid, like theory or utility in practical applications.
> >
> > While this sort of contribution would make the paper stronger, the lack of it is not disqualifying by any means. What stops me from raising my score further is the level of contribution, as other reviewers have concerns about the limited novelty relative to prior works.

---

> > > ### Author Response · Authors · 2023-11-22
> > > **Answer to Reviewer teFB**
> > >
> > > Dear reviewer,
> > >
> > > Thank you very much for your positive review and the score augmentation. We're happy to hear that our responses have been helpful and we are committed to better clarifying our contributions in the future.

---

### Official Review · Reviewer_RCzU · 2023-10-30

**Soundness:** 2 fair
**Presentation:** 2 fair
**Contribution:** 2 fair
**Rating:** 5
**Confidence:** 4

**Summary:**

The paper proposes sparseDiff, there are three main contributions: (1) a noisy model that ensures sparsity is maintained during the diffusion process; (2) a sparse reverse prediction model that performs on a subset of the edge list; (3) an architecture improvement.

**Strengths:**

(1) the development of the model is clear, and the design of each component is demonstrated in full detail.

(2) The architectural design makes sense and convincing.

(3) related works are comprehensive.

**Weaknesses:**

(1) the work is incremental -- the proposed noising process is just taken from DiGress, and the idea of performing sparse prediction at each denoising step is taken from EDGE.

(2) For the second point in (1), the paper claims that there is no guarantee that EDGE's degree-inform sampling always generates such a graph with an exact given degree. However, there is also no guarantee for SparseDiff that the uniform sampling method can learn to reverse the diffusion process. I suggest the author show that there is a proper lower bound of the model when using such reverse distribution.

(3) While SparseDiff advocates sparsity and efficiency, there is no time analysis. I suggest the author should include such discussion theoretically and empirically.

(4) Some baselines are used in some datasets but not in others, can the authors explain why? Some results are not bolded correctly, for example, RBF MMD of SparseDiff in Ego is not significantly better than EDGE. For the table in D.6, DiGress has better RBFMMD than sparseDiff in the protein dataset and EDGE has better FID than sparseDiff in the Ego dataset, however, SparseDiff is bold for both cases.

(5) A larger dataset should be taken into consideration as the author claims scalability in the paper. Validating SparseDiff on QM9 does not support the motivation and the claims.

**Questions:**

See weakness

---

> ### Author Response · Authors · 2023-11-16
> **Answer to Reviewer RCzU**
>
> Dear Reviewer,
>
> Thank you for your thorough review of our paper and the valuable feedback provided. We appreciate your insights and would like to offer clarification on some key issues:
>
> **Regarding the Incrementality of Contributions**
> With SparseDiff, we purposely tried to design an algorithm that is very simple to understand, but that still matches the performance of the best scalable models. We believe that this is useful for future research, as SparseDiff’s simplicity makes it easier to combine with other ideas such as hierarchical generation. Your comments however helped us realize that we did not explain well the implementation challenges associated with our method. Things that look elementary such as sampling edges uniformly or adding noise turned out to be challenging to implement, due to the need to: i) avoid loops for computational efficiency; ii) respect the batching mechanism of PyTorch geometric, which does not use a extra batch size dimension in the tensors, and iii) handle graphs of varying sizes. We will add details regarding these challenges and the way we address them in the manuscript.
>
> **On the Theoretical Guarantee of SparseDiff**
> Establishing a solid theoretical foundation for discrete diffusion models on graphs is a challenging task. In addition to the challenges that are common to all denoising diffusion models, there are some challenges that are specific to graph generation. The biggest one is that graph neural networks (including graph Transformers) are not universal approximators of equivariant functions on graphs. As a result, even with infinite data, memory, and compute, it is not clear that standard denoising diffusion models for graphs can always recover the true data distribution. This is the case for SparseDiff as well as for all other models, which explains why we only provide a limited theoretical analysis. We however believe that the experimental results are convincing in demonstrating the performance of the proposed model.
>
> **Empirical Study on Training Time**
> Thank you for pointing out the analysis regarding efficiency. To better showcase SparseDiff's advantages over DiGress, we present results obtained from training on the large Ego dataset for varying durations. The ensuing table reveals that SparseDiff achieves significantly faster convergence than DiGress. Actually, SparseDiff achieved superior results compared to a DiGress model trained for 4 days, even after only 2 days of training.
>
> |Training Time  |     | 2  |days  |  |  | 4  | days |  |
> |---------- |:----:  | :---: | :---: | :---: | :---: | :---: | :---: | :---: |
> |   Metrics |Deg. |Orbit|Clust.|Spec.|Deg. |Orbit|Clust.|Spec.|
> | DiGress   |0.042|0.185|0.208|0.013|0.033|0.144|0.216|0.011|
> | SparseDiff|0.004|0.053|0.069|0.007|0.002|0.036|0.059|0.004|
>
> **Interpretation of Experimental Results**
> Thank you for noticing the confusing part in experiments and we have diligently rectified the incorrectly bolded numbers. Regarding another concern that baselines for different datasets vary - this is because some results for some datasets are reported by certain baselines, making the baseline models inconsistent across datasets.
>
> **Scalability of the Model**
> We selected Protein and Ego datasets, commonly used in scalable model references, to support our motivation. It's important to note that for these datasets, DiGress struggles to adopt even a very small batch size, making it even more challenging to converge effectively. As an additional result, we attempted training on the largest graph with 1045 nodes from the Facebook dataset and followed SaGess metrics for evaluation. Yet, employing the identical graph for both training and assessing test metrics holds potential risks. Achieving perfect metrics is possible through overfitting to this training graph. The table presents SaGess-RW's outcomes, the best-performing model among the proposed SaGess variants. SaGess forms small graphs, combining them to reach the desired edge count, while SparseDiff generates a single large graph based on the specified node count. Thus SparseDiff matches the 'num nodes' of the real graph, while SaGess approximates 'num edges' more closely. SparseDiff demonstrates scalability to 1000 nodes and excels in performance on single graph datasets, closely aligning with real data statistics, except for the clustering coefficient.
>
> | Model      | num nodes | num edges | num triangles  | num squares | max deg | cluster coef | assort | power law exp | CPL|
> | :-----:     | :----:  | :---: | :---: | :---: | :---: | :---: | :---: | :---: | :---: |
> | Real        | 1045 | 27,755 | 446,846 | 34,098,662 | 1044 | 0.57579 | −0.02543 | 1.28698 | 1.94911 |
> | SaGess      | 1043 | 27,758 | 429,428 | 35,261,545 | 999 | 0.52098 | −0.01607 | 1.29003 | 2.00800 |
> | SparseDiff  | 1045 | 27,763 | 446,819 | 34,095,513 | 1044 | 0.43310 | −0.02536 | 1.28687 | 1.94921 |
>
> Thank you for reviewing our work. We look forward to meeting the requirements in the revised version.

---

### Official Review · Reviewer_hoQ4 · 2023-10-30

**Soundness:** 3 good
**Presentation:** 4 excellent
**Contribution:** 2 fair
**Rating:** 3
**Confidence:** 4

**Summary:**

This paper extends DiGress to
 - use noise model that preserves sparsity during diffusion (although I believe this has been done in DiGress),
 - use a sparse transformer as the denoising model,
 - use a loss function computed on a subset of all pairs of nodes,

enabling training of DiGress on graphs up to 500 nodes. The complexity at sampling time, however, is still $O(N^2)$ due to having to predict the edge existence between every pair of nodes.

**Strengths:**

Diffusion models have shown great potential in graph generation. The issue of scalability in using diffusion model graph generation has been an open problem. This paper addresses that.

**Weaknesses:**

- The contribution of the paper isn't small but the content of the contribution is somewhat trivial. The use of a sparse transformer is an immediate extension to DiGress. Preserving sparsity during diffusion to my knowledge has been done in DiGress.
 - Why the omission of GraphARM and SaGess in the large graph experiments? These two models seem to draw the most parallel comparison for being diffusion-based models with a concern for scalability.
 - Many datasets that contain large graphs have a beta-like distribution: most graphs are small, few graphs are large, making dense training difficult. I'd like to see a study on the benefit of actually including such large graphs in training: can SparseDiff trained on large and small graphs outperform DiGress trained on small graphs, when the task is mostly to generate small graphs?

**Questions:**

See weaknesses

---

> ### Author Response · Authors · 2023-11-16
> **Answer to Reviewer hoQ4**
>
> Dear Reviewer,
>
> Thank you for your thorough review of our paper and your positive feedback about our presentation. We have carefully considered your points and would like to provide clarification on some issues:
>
> **Comparison with DiGress**
> * We would like to clarify the potential confusion regarding the concept of sparsity discussed in both DiGress and our paper. In DiGress, sparsity is preserved throughout the diffusion, ensuring that the number of existing edges remains low compared to all node pairs. DiGress however uses a dense representation of graphs throughout training: it adds noise independently to each node pair, uses all pairs in the message-passing architecture, and computes a loss on all pairs as well. SparseDiff, while also preserving sparsity in noisy graphs, introduces 'computational sparsity' by reducing the noise model’s computational complexity from $O(n^2)$ to $O(m)$. As a result, it enables control over GPU usage that DiGress did not allow, which is key to scalability properties and good performance on large graphs.
> * Besides, SparseDiff is much more than a simple adaptation of DiGress to a sparse library. The design of seemingly basic tasks, such as edge sampling and noise addition, presents significant challenges. These difficulties stem from the need to: i) optimize computational efficiency by avoiding loops; ii) align with PyTorch geometric's batching mechanism, lacking an extra batch size dimension in tensors; and iii) handle graphs of varying sizes. Our manuscript will provide detailed insights into these challenges and our strategies for overcoming them.
>
> **Comparison with GraphARM and SaGess**
> Thank you for proposing those two works for comparison.
> SaGess and our paper both fall under the category of sparse models. However, SaGess conducts experiments using single-graph datasets such as Cora. We believe that using the same graph for training and measuring test metrics is potentially dangerous, as perfect metrics can be obtained by simply overfitting this training graph. GraphARM, while it is a very interesting method, the graphs on why it is benchmarked are unattributed and are not very large. We preferred to showcase performance on molecules and larger graphs, which is why it is not included in the comparison.
>
> For improved comparisons, however, we conducted an additional experiment on the largest graph from the Facebook dataset used by SaGess, which consists of 1045 nodes and 27, 755 undirected edges. It's worth noting that employing the identical graph for both training and evaluating test metrics carries risks, as overfitting the training graph can lead to the attainment of perfect metrics.
> In the table below, we compare SparseDiff with SaGess-RW, the best model among the three proposed SaGess models. Specifically, SaGess generates small graphs and combines them to meet the specified number of edges, while SparseDiff creates a singular large graph based on the given node number. This explains SparseDiff's superiority in the 'num nodes' metric and SaGess's advantage in the 'num edges' metric. Furthermore, SparseDiff is closer to real data statistics, except for the clustering coefficient. This confirms SparseDiff scalability to 1000 nodes and shows its excellent performance on single graph datasets.
>
> | Model      | num nodes | num edges | num triangles  | num squares | max deg | cluster coef | assort | power law exp | CPL|
> | :-----:     | :----:  | :---: | :---: | :---: | :---: | :---: | :---: | :---: | :---: |
> | Real        | 1045 | 27,755 | 446,846 | 34,098,662 | 1044 | 0.57579 | −0.02543 | 1.28698 | 1.94911 |
> | SaGess      | 1043 | 27,758 | 429,428 | 35,261,545 | 999 | 0.52098 | −0.01607 | 1.29003 | 2.00800 |
> | SparseDiff  | 1045 | 27,763 | 446,819 | 34,095,513 | 1044 | 0.43310 | −0.02536 | 1.28687 | 1.94921 |
>
> **Beta Distribution Datasets**
> In common larger graph datasets like Protein and Ego, the graph size variance is not significant, we thus did not specifically consider this scenario during training. On small graph datasets (QM9/Moses/Planar), SparseDiff consistently achieves comparable or even superior results to DiGress. This confirms that SparseDiff maintains excellent performance when training on small graphs. Thus, in this scenario, SparseDiff's performance in small graph generation should be not inferior to DiGress's from an empirical perspective. We are eager to learn more about examples of datasets with beta distribution, as it would enable us to conduct further experiments and validate this in the upcoming version of the paper.
>
> Thank you for reviewing our work. We look forward to meeting the requirements in the revised version. If you have further clarifications or suggestions, please feel free to inform us.

---

### Official Review · Reviewer_h9n8 · 2023-11-06

**Soundness:** 2 fair
**Presentation:** 1 poor
**Contribution:** 1 poor
**Rating:** 3
**Confidence:** 4

**Summary:**

The paper introduces a method that employs sub-graph (edge-wise) sampling and sparse message passing neural networks to enhance the scalability of graph diffusion models during training.

**Strengths:**

The proposed approach uses some techniques to leverage the sparsity of the graphs during training.

**Weaknesses:**

Despite the attempt to leverage sparsity in training, the method's contributions appear to be incremental, lacking significant advancements. Additionally, the graph generation (sampling) is still quadratic which is a bottleneck in such a family of models.

The paper's presentation is incomplete, and the absence of an appendix further hinders clarity. Moreover, several parts of the paper seem to be reiterations of DiGress. There are also some typos and grammatical issues throughout the manuscript.
- Page 4:
  - “Since the noise model is markovian, ~~there~~ the noise does not need”
  - “We denote by k the edge ratio ~~ratio~~ … ”
- page 5:
  - The paragraph “This lemma shows that, in large and sparse graph …” is not clear.
- page 6:
  -  “will define our computational graph ~~$E_c$~~ ~~$G_c$~~ …”
  - “as well as a list of edges denoted $E_c$ …” => “as well as a list of edges denoted by $E_c$ …”
  - “One extra benefit of using a computational graph ~~that~~ with more edges …”
  - “Finally, similarly to standard graph …” => “Finally, similar to the standard graph …”
- Page 8:
  - “We then evaluate or models … ” => “We then evaluate our model … ”

The experimental section lacks a comprehensive empirical study of the model. Comparisons of sampling and training times are important for understanding the model's performance, especially on very large graphs (>1000 nodes), which were not considered to prove the scalability.

**Questions:**

1. The absence of an appendix creates confusion regarding the model architecture and the specific use of PNa and FiLM. Could you provide more details to clarify these aspects?

2. In each call to the model during sampling, as explained in the third paragraph of section 3.4, do you update the edge list?

3. Given that this method is a sparse training form of DiGress, it's unclear why it demonstrates significantly improved generation performance in certain datasets over DiGress. Can you provide insights into the reasons behind this discrepancy?

4. Were the baseline models trained? The reported metrics for EDGE and HiGen on the Ego dataset appear inaccurate. Additionally, the high normalized MMD of Deg. for EDGE and DiGress in this case requires further explanation.

---

> ### Author Response · Authors · 2023-11-17
> **Answer to Reviewer h9n8 - PART 1**
>
> Dear Reviewer,
>
> Thank you very much for your careful review of our paper and the valuable feedback. We have diligently addressed the typo errors you mentioned.
>
> In response to the weaknesses identified in the paper, we would like to provide clarification to better address your concerns:
>
> **Contribution of the Model**
> Your comments on the model contribution helped us realize that the challenging aspects of this work do not stand out well in the manuscript. We believe that the contribution is far from trivial, in the following aspects:
> * While SparseDiff maintains a theoretical quadratic complexity, it presents a practical model that is not only straightforward to comprehend but also readily transferable, offering adaptability to diverse computing environments. The computational complexity of SparseDiff is on average $O(m+ \lambda n^2)$, as the model needs to process both the noisy graph and the query edges. By choosing $\lambda=O(m / n^2)$, it results in a $O(m)$ complexity as opposed to $O(n^2)$ for DiGress.
>
> * SparseDiff is not only about porting DiGress to a sparse library so that it can use sparse computations. Things that look elementary such as sampling edges or adding noise turn out to be challenging to implement. This is due to the need to: i) avoid loops for computational efficiency; ii) respect the batching mechanism of PyTorch geometric, which does not use an extra batch size dimension in the tensors, and iii) handle graphs of varying sizes. We will add details regarding these challenges and the way we address them in the manuscript.
>
> We are revising the text to clarify the challenges that are addressed as well as the contributions that are presented in our work.
>
> **Limitations of the Presentation**
> Thank you very much for pointing out the issues in our presentation. In the revised version, we reorganized the paper structure to highlight better our contribution. This involves streamlining the introduction within the Method section and emphasizing the distinctions from DiGress, as well as a clearer description of the actual challenges in scaling up diffusion-based graph generation models.
>
> **Empirical Study on Sampling and Training Time**
> We provide an empirical result here to better demonstrate the advantages of SparseDiff compared with DiGress. We report the result after being trained for different times with the Ego dataset. The following table demonstrates that SparseDiff converges much faster than DiGress on this large dataset. SparseDiff achieves superior results compared to a DiGress model trained for 4 days, even after only 2 days of training.
>
> |Training Time  |     | 2  |days  |  |  | 4  | days |  |
> |---------- |:----:  | :---: | :---: | :---: | :---: | :---: | :---: | :---: |
> |   Metrics |Deg. |Orbit|Clust.|Spec.|Deg. |Orbit|Clust.|Spec.|
> | DiGress   |0.042|0.185|0.208|0.013|0.033|0.144|0.216|0.011|
> | SparseDiff|0.004|0.053|0.069|0.007|0.002|0.036|0.059|0.004|
>
> Finally, we acknowledge that our approach introduces higher time complexity during the sampling process. Our work focused on the ability to train denoising diffusion models without assuming specific characteristics of the data distribution, and we did not try the various techniques that exist to optimize sampling time.
>
> **Training on Larger Graphs**
> We attempted training on the largest graph with 1045 nodes from the Facebook dataset, following the setting in SaGess. We also evaluated SparseDiff using SaGess metrics as a reference.  However, using the same graph for both training and evaluating test metrics is potentially risky, as achieving perfect metrics can be accomplished through overfitting to this training graph.
> In the table below, we report SaGess-RW, which yields the best results among the three proposed SaGess models. For comparison, it is noteworthy that SaGess generates small graphs and concatenates them until reaching the required number of edges, whereas SparseDiff generates a single large graph based on the given node count. This accounts for the advantage of SparseDiff in the 'num nodes' metric and the advantage of SaGess in the 'num edges' metric. Additionally, SparseDiff aligns more closely with real data statistics, except for the clustering coefficient, which not only validates its scalability up to 1000 nodes but also underscores its high performance on such single-graph datasets.
>
> | Model      | num nodes | num edges | num triangles  | num squares | max deg | cluster coef | assort | power law exp | CPL|
> | :-----:     | :----:  | :---: | :---: | :---: | :---: | :---: | :---: | :---: | :---: |
> | Real        | 1045 | 27,755 | 446,846 | 34,098,662 | 1044 | 0.57579 | −0.02543 | 1.28698 | 1.94911 |
> | SaGess      | 1043 | 27,758 | 429,428 | 35,261,545 | 999 | 0.52098 | −0.01607 | 1.29003 | 2.00800 |
> | SparseDiff  | 1045 | 27,763 | 446,819 | 34,095,513 | 1044 | 0.43310 | −0.02536 | 1.28687 | 1.94921 |

---

> ### Author Response · Authors · 2023-11-17
> **Answer to Reviewer h9n8 - PART 2**
>
> In response to your specific questions, we provide the following answers:
>
> **Specific Usage of PNA and FiLM**
> We appreciate very much your careful reading. In the revised version, we will provide a more detailed explanation to address any confusion regarding the model architecture and the FiLM and PNA layers.
> * FiLM layer: We utilize FiLM to combine features at different scales. Specifically, given two features $M_1$ and $M_2$, and trainable parameters $W_1$ and $W_2$, the FiLM layer output is calculated as $M_1 W_1 + (M_1 W_2) \odot M_2 + M_2$. As an illustration, within the convolutional layer, the graph feature $M_2$ is integrated with edge features $M_1$ to enhance predictions.
> * PNA layer: We use the PNA layer as a specialized pooling layer to obtain information from different dimensions of a specific feature. Given the feature $X$, $PNA(X) = {cat}({max}(X), min(X),  mean(X), std(X)) W$. For example, node features $X$ are forwarded to a PNA layer for extracting global information across different scales, which is subsequently added to the graph feature to enhance its representation.
> * Model architecture: Our model architecture follows a similar yet much lighter version of DiGress. Its Transformer layers use the message-passing attention mechanism, rather than dense attention as in DiGress. It incorporates FiLM layers to harness distinct levels of information. For instance, FiLM helps leverage global graph features into both node and edge features. Simultaneously, the PNA layer is employed to conduct pooling operations on node, edge, and graph features, respectively. Besides, the model has been lightened by its Transformer layers using the message-passing attention mechanism, rather than dense attention as in DiGress. More details can be found at: https://anonymous.4open.science/r/SparseDiff-B861/.
>
> **Updating the Edge List**
> We first consider all n(n-1)/2 indices representing pairs of nodes and randomly permute them. We cut the resulting array into equal-sized chunks that represent the query graph at each iteration. We then iterate over these blocks, while keeping the noisy graph $G^t$ fixed. This is explicitly stated in the revised manuscript.
>
> **Performance Improvement of SparseDiff**
> We agree that on small graphs, there should be no benefits of using sparse operations. We believe that the superior performance of SparseDiff over the version of DiGress with charges is due to small variations in the architecture (as explained above), but we were surprised by the results as well.
>
> **Ego Dataset Experiment Metrics**
> We are very sorry for this mistaken information that we wrongly reported the SBM results instead of EDGE. The inaccuracies in this section have been corrected, and the revised results compared to SparseDiff are as follows.
>
> | Model      | Degree | Clustering | Orbit  | Spectre | RBF |
> | :-----:     | :----:  | :---: | :---: | :---: | :---: |
> | EDGE        | 290 | 17.3 | 4.3 | - | $4.0$ |
> | HiGen       | 236 | 0.3|3.2| - | $3.7$ |
> | SparseDiff  | $9.5_{\pm{3.5}}$ | $5.4_{\pm{.2}}$ | $2.5_{\pm{.1}}$ | $3.6_{\pm{1.1}}$ | $3.9_{\pm{1.0}}$ |
>
> Regarding the high result in Deg MMD, we were also surprised to observe that most methods performed poorly on this seemingly simple metric. As the reported results are from previous papers, we unfortunately do not have access to the degree distribution of the samples for the baselines, and cannot tell where the bad metrics come from.
>
> Thank you for your review, and we look forward to meeting the requirements in the revised version. If you have further clarifications or suggestions, please inform us.

---

### Author Response · Authors · 2023-11-16
**Response to All Reviewers**

Dear Reviewers,

We greatly appreciate your constructive feedback and insightful suggestions, which are very helpful in refining our manuscript. We appreciate that you recognized sparsity as an important feature for the next generation of generative models for graphs.

Allow us to highlight the key contributions of SparseDiff:
* Our empirical experiments underscore the model's outstanding performance across diverse datasets. Additional experiments (added in the revised manuscript) further validate SparseDiff's superior convergence speed on large graph datasets compared to DiGress.
* From a practical standpoint, SparseDiff's ease of control stands out, requiring a straightforward adjustment of the parameter $\lambda$ for flexible GPU utilization. Moreover, we believe that our implementation can be combined with ideas proposed in other works (such as hierarchical generation) in order to scale to even larger graphs.
* Despite SparseDiff's apparent simplicity, it involves significant and intricate algorithm design, which makes the proposed contribution relevant and certainly non-trivial in our opinion.

In response to your valuable feedback regarding the presentation, we have made the following main revisions to the manuscript:
* The introduction and related work sections have been refined to improve clarity.
* The method section has undergone substantial improvements, including the addition of a detailed description of the training algorithm in the main body. We have also optimized the structure of the 'sparsity-preserving noise model' section, emphasizing the distinctions between SparseDiff and DiGress.
* The experimental section now includes supplementary results on training convergence time, highlighting SparseDiff's advantages in efficiency. We have also provided a more comprehensive explanation of the motivation behind using 5-fold sampling to increase result stability, encouraging the adoption of a standardized approach within the research community.

We genuinely appreciate the time and effort you dedicated to reviewing our work. Your continued guidance is invaluable, and we are committed to addressing any further concerns you may have. More detailed responses to each reviewer are provided below.

---

### Meta-Review · Area_Chair_yNpe · 2023-12-06

**Metareview:**

The paper focused on enhancing the scalability of graph diffusion models and presented SparseDiff, a novel approach by combining sub-graph sampling with sparse message-passing neural networks, which is a pertinent topic in the field.

While SparseDiff addresses an important problem in graph generation, the incremental nature of its contributions, coupled with significant issues in presentation and experimental validation, do not justify its acceptance. The paper needs substantial improvements in clarity, theoretical grounding, and empirical evidence to meet the conference's standards.

**Justification For Why Not Higher Score:**

- The primary concern is the incremental nature of the contributions, with the paper lacking substantial advancements over existing models like DiGress.
- The paper's presentation is notably weak, suffering from clarity issues due to incomplete information, absence of an appendix, and several grammatical errors. There's also a lack of clarity in the methodology description, with fragmented explanations and an absence of a clear, step-by-step algorithmic representation.
- The experimental section lacks depth, particularly in demonstrating the model's performance on very large graphs, a critical aspect for proving scalability.

While the authors showcased improvements during rebuttal, it is evident that this paper needs substantial rewriting and cannot be accepted in current shape.

**Justification For Why Not Lower Score:**

N/A

---

### Decision · Program_Chairs · 2024-01-16

Reject